# Profiling Phospholipids within Atlantic Salmon *Salmo salar* with Regards to a Novel Terrestrial Omega-3 Oil Source

**DOI:** 10.3390/metabo12090851

**Published:** 2022-09-10

**Authors:** Richard Broughton, Douglas R. Tocher, Johnathan A. Napier, Mónica B. Betancor

**Affiliations:** 1Institute of Aquaculture, Faculty of Natural Sciences, University of Stirling, Stirling FK9 4LA, UK; 2Guangdong Provincial Key Laboratory of Marine Biotechnology, Institute of Marine Sciences, Shantou University, Shantou 515063, China; 3Department of Plant Sciences, Rothamsted Research, Harpenden AL5 2JQ, UK

**Keywords:** lipidomics, phospholipids, salmon, aquaculture, GM, novel feeds

## Abstract

The development and inclusion of novel oils derived from genetically modified (GM) oilseeds into aquafeeds, to supplement and supplant current terrestrial oilseeds, as well as fish oils, warrants a more thorough investigation into lipid biochemical alterations within finfish species, such as Atlantic salmon. Five tissues were examined across two harvesting timepoints to establish whether lipid isomeric alterations could be detected between a standard commercial diet versus a diet that incorporated the long-chain polyunsaturated fatty acids (LC-PUFA), EPA (eicosapentaenoic acid), and DHA (docosahexaenoic acid), derived from the GM oilseed *Camelina sativa*. Tissue-dependent trends were detected, indicating that certain organs, such as the brain, have a basal limit to LC-PUFA incorporation, though enrichment of these fatty acids is possible. Lipid acyl alterations, as well as putative stereospecific numbering (sn) isomer alterations, were also detected, providing evidence that GM oils may modify lipid structure, with lipids of interest providing a set of targeted markers by which lipid alterations can be monitored across various novel diets.

## 1. Introduction

Aquaculture is tasked with fulfilling the demands of a growing population, owing to constraints on the capacity of traditional wild capture fisheries [1]. In the future, the role of fisheries, as well as many other food production systems, will likely be impacted by climate change, though the impact is likely to not be distributed equally across countries and communities [2]. Therefore, to meet the demands of a growing population, aquaculture will need to provide the additional capacity by which sustainable and nutritious seafood is produced. Regarding human nutrition, fish consumption is advised on a weekly, or bi-weekly basis, based on the health benefits provided by the high levels of protein, micronutrients, and omega-3 long-chain polyunsaturated fatty acids (LC-PUFA) [3,4]. It is the latter, however, the omega-3 fatty acids that are present in significant amounts, which consumers particularly associate with seafood. Owing to the biotrophic accumulation of these fatty acids, aquaculture feeds need to contain either marine-derived fishmeal and/or fish oil to supply finfish with omega-3 LC-PUFA. Vegetable oils, such as rapeseed, are typically used to supply and increase the energy content of feeds and can provide fatty acids including 18:3n-3, however terrestrial plant oils lack LC-PUFA. Presently other alternative oil sources rich in marine fatty acids are being investigated (e.g., oils derived from heterotrophic microalgae), but scalability and price are still a constraint, though are improving and are already used within the industry.

Therefore, nutritional quality for consumers and issues related to fish health are potential issues in aquaculture [5], owing to the growing demand to produce fish rich in omega-3 LC-PUFA with constrained lipid supplies. Omega-3 LC-PUFA containing lipids can be obtained from lower trophic levels, including krill, copepods, and algae. However, depleting the natural stocks of these organisms could alter the ecosystem balance, resulting in potentially disruptive ecosystem alteration [6,7]. Therefore, novel solutions are required to address the need for these high-value lipids. One solution is the adoption of genetically modified (GM) oil seed crops, capable of elongating and desaturating fatty acids beyond the standard terrestrial lipids [8,9,10], which has been undertaken within *Brassica napus* and *Camelina sativa* [11], though only *Brassica napus* varieties are currently commercially offered. Within the present study, an oil derived from *Camelina sativa*, rich in both 20:5n-3 (eicosapentaenoic acid, EPA) and 22:6n-3 (docosahexaenoic acid, DHA) was trialled to assess its impact on the lipid profiles of several tissues of Atlantic salmon (*Salmo salar*) in comparison to a standard commercial-like vegetable oil/fish oil blend (control). The assessment of lipid composition at two different timepoints was used to ascertain the plasticity of lipids, with the first timepoint 14 weeks prior to the final sampling. Little work has been conducted applying lipidomic techniques to the assessment of novel feeds, and whilst lipid composition may be impacted by factors such as season and fecundity, assessment of diets within commercially relevant conditions was of initial interest and provides a basis for further exploration of more fundamental alterations to lipid biochemistry.

The use of oils derived from GM sources may allow for the replacement of fish oil, and traditional vegetable oils, however, owing to the synthetic biology approach, the structure of the lipids in GM-derived oils may differ from those produced natively, through both de novo and dietary incorporation. Alterations may stem from the combinations of desaturases and elongases used within the host plants [11,12], as well as the potential for the host organism itself to impact lipid biosynthesis. This approach has resulted, in certain instances, in acyl rearrangement within triacylglycerols (TAG). Both NMR and lipase studies have generally concluded that DHA preferentially occupies the sn-2 position in fish oils [13,14,15,16], with EPA usually being equally split or being more likely in the outer sn-1/3 positions, while it has been shown that DHA resides preferentially in the outer sn-1/3 positions in TAGs in engineered systems [9,17], while EPA positioning seems less affected. The relative proportions of DHA in single cell oils such as *Schizochytrium* sp. and *Crypthecodinium* appear to be enriched within the outer sn-1/3 positions [18], though these are calculated values rather than directly observed. However, few comparisons of other fatty acids in fish oils and GM oils have been conducted, possibly because ^13^C NMR, the predominant method for acyl sn determination has no distinct carboxylic functional group peak for certain fatty acids, or peaks represent a mixture of fatty acids [19], therefore warranting a lipase-based approach. This regiospecificity has also now been modified within TAGs for specific fatty acids, for the desired market [20]. Such structural differences are generally not evident using standard lipid analysis, such as GC, whereby the intact lipid structure is usually destroyed, or LC-MS with standard reverse phase conditions that are unlikely to resolve sn positional isomers or D/L stereoisomers. However, the synthetic construction of TAG, the main dietary lipid within aquafeeds, inherently involves multiple elongation and desaturation steps, potentially with several lipid substrates, and may contain subtle variations not commonly found within traditional lipid sources. This may impact both the digestion and subsequent utilisation of the fatty acids by various fish tissues, resulting in a unique signal, detectable as alterations within the isomer ratios. Detection of isomeric variations in relation to GM oils has received little attention and potentially may impact fish health through sn specific lipase action for eicosanoid production, and incorporation into LC-PUFA-dependent tissues, such as the brain and eye.

To that end, a preliminary screening protocol based on supercritical fluid chromatography SFC-MS/MS was utilised to determine and characterise lipids, which distinguished fish that had been fed control and GM oil-supplemented feeds, with candidate compounds further explored using non-standard, targeted LC-MS/MS methodology, which was designed to resolve phospholipid isomers.

## 2. Materials and Methods

### 2.1. Materials

Acetonitrile, methanol, and isopropanol were supplied by Rathburn Chemicals (Walkerburn, UK), and formic acid, water, and ammonium formate were purchased from Merck (Dorset, UK), with all solvents being LC-MS grade. CO_2_ (99.8% industrial grade) was purchased from BOC (Grangemouth, UK). HPLC grade methyl-tert-butyl ether (MTBE) and chloroform were purchased from Fisher Scientific, and LC-MS grade sample vials were purchased from Waters (Waters, Milford, MA, USA).

### 2.2. Fish Growth Conditions and Sampling

All procedures were conducted in compliance with the Animals Scientific Procedures Act 1986 (Home Office Code of Practice) and in accordance with the regulations set forward by the Directive 2010/63/EU of the European Parliament and of the Council of 22 September 2010 on the protection of animals used for scientific purposes. Additionally, the experimental protocol was reviewed and approved by the Animal Welfare and Ethical Review Board at the University of Stirling (AWERB(16-17)83).

A total of 900 post-smolt, mixed sex, Atlantic salmon (*Salmo salar* L.) with an average initial body weight of 187.2 ± 1.9 g (mean ± SD) were distributed into six 5 m seawater floating pens (150 fish per pen) and fed one of two experimental diets in triplicate from week 25, 2018 to week 10, 2019 with the average fish weight at the intermediate point being 1941.8 ± 156.3 g, and the final average weight being 3510.8 g ± 158.3 (mean ± SD). The intermediate timepoint recorded an average weight of 2071 g ± 97.3 for the control diet, 1812 g ± 46.3 for the GM diet, and for the final timepoint, 3601.9 g ± 307.6 for the control diet and 3108.3 g ± 87.0 for the GM diet, with no significant difference between dietary treatments. The diets were isolipidic (36%) and isoproteic (36%) and formulated to contain either a 2:1 (*v*/*v*) blend of vegetable (rapeseed) and fish oil (control diet), or an oil derived from the genetically modified oilseed *Camelina sativa*, rich in both EPA and DHA (GM diet) (Table 1). Six fish per pen were humanely euthanised by anaesthetic overdose of metacaine sulphonate (>150 mg/L) at two timepoints, week 48 2018 (intermediate), and week 10 2019 (final). Five tissues including the brain, eye, gill, intestine, and liver were collected at each timepoint and immediately frozen at −70 °C prior to lipid extraction.

### 2.3. Lipid Extraction

Lipids were extracted from pooled tissues, with three pooled fish per biological replicate, with two pools taken from each pen and were treated as six biological replicates per diet (*n* = 6). Lipids were extracted using the MTBE method [21]. Samples were kept on ice during extraction, with gill and intestine samples manually chopped prior to homogenisation. Pre-cut or intact tissues were weighed into 50 mL test tubes, 3 mL of 90% *v/v* ice-cold methanol per gram of tissue was added and samples were homogenised using an Ultra-Turrax tissue disrupter (T25, IKA, Darmstadt, Germany). Once homogenous, the total sample volume was measured, and a volume equating to 200 mg of sample was taken into a glass test tube and 5 mL of MTBE added, samples vortexed, and left on ice for 5 min. Then, 1.25 mL of water was added, samples vortexed, and centrifuged at 800 g for 5 min at room temperature to allow the solvent system to separate. The upper phase was taken into a clean glass tube, and a second extraction was carried out on the lower phase using 2 mL of MTBE/methanol/water (10:3:2.5, *v*/*v*/*v*). After centrifugation, the second upper phase extract was pooled with the first, and the total extract was dried under nitrogen. Total lipid extracts were then weighed, and a stock solution of 10 mg/mL in chloroform/methanol (2:1, *v*/*v*) was made. QC samples were made by mixing equal volumes of both control and GM extracts together, to create a mixed sample that could be reinjected several times throughout a run. The stock concentration was used for positive mode, whilst a 2.5 times concentration was used for negative mode injections.

### 2.4. Untargeted Supercritical Fluid Chromatography (SFC)-MS/MS

Untargeted SFC-MS/MS was carried out as described previously [22], with polar lipids studied separately from neutral lipids. SFC separation was carried out on an Acquity UPC^2^ system (Waters, Milford, MA, USA), which used CO_2_ as solvent A, and methanol/acetonitrile/water (75:20:5, *v/v/v*) containing 0.1% (*v/v*) formic acid and 0.15% ammonium formate (*w/v*) as co-solvent B. The makeup solvent comprised of methanol/isopropanol (80:20, *v/v*) containing 0.1% (*w*/*v*) ammonium formate. A 5 cm Viridis C18 HSS column (50 × 2.1 mm, 1.8 µm particle size; Waters, Milford, MA, USA) was used with neutral lipid analysis decoupled from polar lipid analysis by diverting column eluent to waste prior to the elution of polar lipids. SFC gradient elution comprised of solvent B being held at 2% for 2.5 min, then increased to 64% at 18.7 min, held at 64% solvent B until 24 min, then decreased to 2% at 27.5 min, and held at 2% for a final run time of 29.5 min. The back pressure regulator was set to 1500 psi, the flow rate was set at 1.2 mL/min, and the makeup solvent flow rate set to 0.25 mL/min, with the column temperature set at 52.5 °C. Two µL of the sample was injected on-column.

The UPC2 system was attached to a Xevo G2-XS Q-TOF mass spectrometer (Waters, Milford, MA, USA), which was operated in both positive and negative ESI mode. The capillary voltages were set to 3 kV, with the sampling cone voltage set at 28 V, the source offset set to 80, and the source and desolvation temperatures set at 120 and 300 °C, respectively. The cone and desolvation gas flows were set to 50 and 1000 L/h, respectively. The system was operated in MS^e^ mode, with the scan rate set between 200 and 1600 Da at 0.25 sec/scan, with a low energy collision energy of 2 V and a ramped voltage of 28–40 V for the high energy scan. The lock mass compound was leu-enkephalin, which was infused at a rate of 10 µL/min during the run to correct for mass deviations.

### 2.5. Semi-targeted Reverse Phase LC-MS/MS

Semi-targeted analysis was conducted on a Waters I class UPLC, attached to a Xevo TQ-S, based on a method by Nakanishi et al. [23]. The reverse phase method utilised two 10 cm Aquity BEH C18 columns (100 × 2.1 mm, 1.7 µm) in tandem, and four solvents. Solvent A comprised acetonitrile/water (90:10, v/v) with 0.1% ammonium formate (*w/v*), solvent B comprised of acetonitrile/methanol/isopropyl alcohol/water (47.5:45:2.5:5, *v/v/v/v*) with 0.1% ammonium formate (*w/v*), solvent C comprised of acetonitrile/methanol/isopropyl alcohol (40:45:15, *v/v/v*) containing 0.1% ammonium formate (*w/v*), and solvent D comprised 100% isopropyl alcohol. The flow rate was set at 0.33 mL/min, with a column temperature of 65 °C. The system gradient used is shown in Table 2.

The Xevo TQ-S was used in data-dependent acquisition mode, with an inclusion list for fragmentation based upon the lipids identified using untargeted QTOF methodology. The system was run in negative mode with a capillary voltage of 3 kV, a cone voltage of 28 V, a source offset of 80, a source and desolvation temperature of 130 and 280 °C respectively, and a cone and desolvation gas flow of 150 and 1000 L/h, respectively. The survey scan was adjusted based on the target lipids and was set to 680–970 Da with a scan time of 0.35 s. Fragmentation of lipids was carried out through the use of an inclusion list, which utilised a ScanWave DS fragmentation of precursor ions. The fragmentation mass range was set from 100–950 Da and scanned at 5000 amu/sec.

### 2.6. Data Analysis

The UPC2 and Xevo G2-XS QTOF were operated using Masslynx v4.2, whereas the Xevo TQ-S was operated using Masslynx V4.1. Xevo G2-XS QTOF data were analysed using Progenesis QI v 3.0 (Nonlinear Dynamics, London, UK), with multivariate OPLS-DA analysis run using Simca-P v12.0 (Umetrics, Umeå, Sweden). Each tissue type and timepoint was analysed separately, comparing the GM diet against the control diet, with data normalised using the all-ions approach. Progenesis parameters were set as; ANOVA score < 0.05, a fold change of >1.2, and multiple testing values (Q values) of <0.05 were used to filter results. Multivariate analysis was then used to discriminate between the control and GM groups, using OPLS-DA to determine compounds of significance, with a w (1) score higher than ±0.04, and a p (corr) score greater than ±0.6. QC samples were assessed to establish whether these clustered centrally between the two treatments. These compounds were reimported back into Progenesis for lipid identification, based on retention time, accurate mass, and fragmentation. Data were then exported to Excel, with the positive and negative mode data then merged to create a unified dataset.

The selected compounds from each tissue type and timepoint were converted into their negative ion form, with these compounds forming the basis of the m/z range to be chosen for the LC-MS/MS approach, as well as the data-dependent acquisition fragmentation inclusion list. Those lipids which were identified within the tissue or specific timepoint by QTOF were integrated using Masslynx, with the fragmentation profiles interpreted to deduce the lipid acyl composition. Integrated profiles were exported to Excel for further analysis, which included percentage calculations of higher-level lipid structure, e.g., PC 42:10, and ratio calculations of putative sn isomers. Data relating to lipid structure and statistical values are given in Appendix A. Data were multiple test corrected using the Benjamini-Hochberg procedure, and those below the cut-off point are highlighted in red, as shown in the Appendix A.

## 3. Results

### 3.1. Tissue Responses to GM and Control Diets

Five tissues were examined in the present study, two of which, the brain and eye, were found to be more conservative in terms of fold change when the average fold change across polar lipid classes was considered (Figure 1). The largest number of lipid alterations were observed within phosphatidylcholine (PC) and phosphatidylethanolamine (PE), but these were generally limited to around, on average, 2-fold in both these tissues. Average fold changes in PE were generally much greater for the other three tissues, ranging from 3 to 5-fold on average. On the other hand, PC was more conservative, only extending up to a change of 3-fold on average. Similarly, phosphatidylglycerol (PG) generally showed higher fold variances in the intestine and liver compared with the brain as did phosphatidylinositol (PI), which was tightly controlled in the brain, showing similar fold changes across timepoints, whereas the other tissues, especially liver, generally showed both greater variance and higher average fold changes. Ceramides and the more tissue-specific glycosylated sphingolipids and galactolipids showed little in the way of overall tissue trends, with average fold changes close to 2, and tightly controlled variances. A few cardiolipins were also detected, although the majority appeared to respond to the control diet, with the greatest diversity detected within the eye at the final timepoint. Cardiolipins were not summarised further owing to the small number detected.

With regards to differential accumulation of lipid class contents between diets, whilst various individual lipids were observed within the different tissues, a few general trends were apparent. The sum of the lipids comprising the main phospholipid classes, PC and PE, demonstrated quite marked alterations in the gill at both timepoints (Figure 2), with these two classes generally being elevated when fed the control diet. The intestine and the earlier liver timepoint generally showed increased PC in fish fed the control diet, whereas the final brain timepoint, on average, showed PC was increased in fish fed the GM diet. However, PE displayed a relatively equal split for all tissues except the gill. The hexose ceramides, which also included the galactolipids monogalactosyl diacylglycerol (MGDG) to increase the span of the group, were found to be predominantly increased by the GM diet within the brain, as were the limited number of ceramides. This appeared in contrast to the eye where, in general, fish fed the control diet showed increases in tissue-specific glycosylated lipids, distinct from the brain (Figure 3). Diet had little impact on ceramide contents in the final gill and intermediate intestine samples, though both the intermediate gill and liver showed increased ceramides in fish fed the GM diet.

As above, PG again seemed to be selectively increased within brain tissue, as well as the intermediate liver sampling point, however, this was reversed in the final liver sample, which showed that PG tended to be increased in fish fed the control diet. This was also found to be the case within the final gill and intestine samples. Finally, for both PI and phosphatidylserine (PS), the brain seemed to be most affected when fed the GM diet, as again, PI was almost exclusively increased within this tissue. In contrast, PI in the eye appeared to be increased in fish fed the control diet, at least at the intermediate timepoint. Similar trends in PI were observed in the gill, intestine, and liver at the same timepoint, though the liver demonstrated a much wider variation, possibly owing to the relatively low abundances detected within the tissue. Significant levels of PS were only detected in the eye and appeared to increase in fish fed the control diet. It should be noted, however, that individual lipids were not always detected or studied in each tissue, although general trends present within both the tissue and at different timepoints appear to be evident.

When the individual lipids are deconstructed into both their carbon and double bond numbers (Figure 4), several trends become evident. With regards to carbon number, brain tissue illustrates that there appears to be a lower limit of lipid modification, which tightens with brain maturity. On average, the brain appears to defy the trends of the other tissues, with all even carbon number lipids being generally elevated in response to the GM diet, which is especially evident at the final timepoint. The gill, and to some extent the earlier intestine timepoint, bucked this trend somewhat, being more likely to show increased fold change with a higher carbon number in fish fed the control diet. Trends that appeared relatively consistently included the majority of odd chain lipids being increased in fish fed the control diet and, with the exception of brain tissue, 30, 32, 34, and, to some extent, 36 carbon lipids were also elevated in fish fed the control diet. Excluding gill, the larger 42 carbon lipids appeared to be elevated with the GM diet, with 44 carbon lipids also favoured within the liver.

Regarding the number of double bonds, the control diet appeared to result in increased mono- and di-unsaturated lipids and, to a lesser extent, saturated lipids. The brain once again showed the opposite trend for monounsaturated lipids, most notably by the final timepoint, showing elevated levels in fish fed the GM diet. The brain generally showed a tight spread of fold changes, as shown with the earlier 2-fold change on average, and usually in a predominant direction. At the intermediate sampling point, the brain conformed to the general pattern shown in the other tissues, with 6 and 7 double bond-containing lipids increased in fish fed the control diet. In fish fed the GM diet, the brain generally showed increased levels of the majority of other double bond numbers, whereas the eye showed increased levels of 8 and 9 double bond-containing lipids. Regarding overall trends for more unsaturated lipids, 4, 9 and to a lesser extent 10 double bond lipids on average appeared to be favoured in fish fed the GM diet, though not as consistently as with mono- and di-unsaturated lipids in salmon fed the control diet. Gill and intestine appeared to show much tighter control in terms of fold changes and often were the tissues most reliably modulated by the control diet, with lipids containing 3, 5, 6, and 7 double bonds more consistently increased by the control diet.

This consensus between both tissue and timepoints was present for a large number of individual lipids, with lipid responses consistent across a diet (Figure 3). Where no response was observed for a specific tissue or timepoint it could be assumed that either the trend would be in the same direction or was not observed due to the change being too small and/or not significant. These consistent patterns were most evident across PC and PE, with increased levels in fish fed the GM diet being seen in PE 42:9, 38:4, and 36:4, as well as PC 36:4 and 38:4, for example. Counter trends existed in PE 40:7 and ether lipids such as PE O-40:8, as well as PC 36:6 and 40:7, which saw consistent elevation by the control diet. It also appears that PC and choline-containing lipids had more individual species modulated by the control diet, approximately 78 versus 53 for the GM diet. Of note, the brain in general tended not to show much response to the control diet, as evidenced by a greater proportion of positive fold changes, representing proportionally higher lipid elevation brought about by the GM diet. This can be seen in Figure 3, especially within the PC class, where lipid species elevated by the control diet are not seen to be modulated within the brain, with inter-tissue trends only seen in lipid species that were elevated by the GM diet, indicating some potential form of lipid compositional baseline or threshold, which can only be increased, but not decreased. Gill on the other hand tended to lack the wealth of individual lipids which were modified in other tissues, although was generally starkly contrasted to the brain, being more sensitive to the control diet and rarely showing individual lipids upregulated by the GM diet.

These trends across tissues also highlight the possibility that certain trends may exist across entire lipid classes, even in the absence of comparable data across both tissue and time. For example, with PG there appeared to be a diet-related overall trend shared across individual tissues or timepoints, for example, the final timepoints of gill, intestine, and liver, even though the same PGs are not always detected across different tissues. There are also contra directional changes for both individual and entire lipid species indicating that, in fact, a whole lipid class may be modulated within a tissue in the same manner by dietary conditions. These contra directional changes occur to a limited extent, being evident in the liver across time, predominantly in the PG lipid class, although occasionally occurring in PC, for example, PC 42:7. Inter tissue variations also exist, for example, PE O-40:7, PE 40:6, PG 36:5 and PC 42:7.

### 3.2. Isomeric Analysis

Those lipids which demonstrated significant change within the QTOF dataset (Figure 3) were selectively targeted using reverse phase LC-MS/MS methodology. Owing to the selective nature of the method, fully comprehensive profiling of each lipid class was not possible. Therefore, predominantly PC and PE were studied, owing to their amenable fragmentation profiles in negative mode, though only those lipids detected in the initial analysis above could be analysed in further detail. It was demonstrated, however, that the reverse phase method was capable of resolving potential sn positional isomers (Figure 5). As illustrated in Figure 5, acyl ion fragments were clearly evident in negative mode, however, their use in determining the sn configuration was hampered by the non-reproducible ion ratios that derive from the acyl fragments. Therefore, the isomeric configurations are currently uncertain and were designated as putative sn isomers, with these lipid isomers given a/b/c suffixes.

Once lipids were separated into individual acyl isomers, those with two or more isomers were taken forward, with several acyl trends determined (Figure 6). On average across all tissues, and within the PC lipid pool, and most evident for the fold change data, lipids containing 14 and 16 acyl groups were more likely to be associated with the control diet, that is, complex lipids were more likely to contain one of these acyl groups in one of the two acyl positions (Figure 6A). Conversely, lipids containing acyl groups of 20 and 24 carbons in length were more likely to be associated with the GM diet, though 18 and 22 carbon acyl-containing lipids, on average, also tended to be more likely to be increased by the GM diet. No obvious discernible pattern was determined within the PE lipid pool (Figure 6B). With regards to double bond number, across the PC and PE (Figure 6C/D) lipid pools within the selected lipids and across all tissues, monounsaturated acyl groups were more likely to be attached to complex lipids and increased in salmon fed the control diet. Contrary to this, 3 and 4 double bond-containing acyl groups incorporated into complex lipids were more likely to be influenced by the GM diet, though only the PC lipid pool saw a general increase of 5 double bond-containing acyl groups in fish fed the GM diet.

When the PC and PE pools were combined but split on the basis of tissue and timepoint (Figure 7), only those that contained a reliable number of lipids were analysed, with these constituting acyl groups containing 16 to 22 carbons. Sixteen carbon acyl groups were found to be marginally increased in fish fed the GM diet within the liver, though only when looking at percentage data. Acyl groups comprising 18 carbons tended to be preferentially incorporated within complex lipids in fish fed the control diet, though only evident in percentage data, with gill exhibiting the greatest response. Fold data clearly showed a preference for 20 carbon acyl chains being increased by the GM diet, with the strongest effects observed in the brain and eye, while gill showed no preferential regulation. Acyl groups containing 22 carbons were less clear cut, although fold data demonstrated that intestine and liver, as well as the final eye timepoint, responded with marginally increased inclusion in complex lipids when fed the GM diet. In contrast, gill generally presented increased incorporation of these acyl groups when fed the control diet.

Regarding the levels of unsaturation for the combined PC and PE pools (Figure 8), fold change data tended to give a clearer indication of patterns within tissues and timepoints. Saturated acyl groups appeared to partially be more likely to be incorporated into complex lipids, and were increased by the GM diet, whilst the final timepoints for both brain and eye, containing di-unsaturated acyl groups, also experienced similarly increased levels when expressed as fold change in response to the GM diet. Acyl groups containing 3 and 4 double bonds were more consistently increased by the GM diet, with acyl groups containing 5 and 6 double bonds also marginally elevated. As evidenced by the fold change data, a countertrend was evident within the gill at both timepoints, indicating that the inverse was true; that the control diet increased the production of lipids containing acyl groups containing 5 or 6 double bonds. As seen previously, mono-unsaturated acyl groups tended to be increased by the control diet.

The limited number of significantly modified PC and PE lipids in the brain stemmed from the initial QTOF analysis, and again, individual acyl isomers were muted in terms of fold change, and more so in terms of percentage compositional shifts within the carbon and double bond classes, when compared with the other tissues (Appendix A). On the other end of the spectrum, unsurprisingly, the liver and intestine showed the greatest variation with respect to percentage composition, although the eye at the intermediate timepoint also demonstrated a surprisingly large variation. The liver intermediate timepoint demonstrated the greatest fold variance, with the gill intermediate timepoint showing the second highest variance. The number and distribution of acyl isomers which constitute a lipid class structure, e.g., PC 36:2, tend to be relatively consistent across tissues and timepoints, with the caveat that a total data set encompassing all observed acyl isomers is not available for all tissues and timepoints, owing to the technical limitations of the instrumentation. However, that noted, several tissue or timepoint acyl alterations are evident, with noticeable lipids being PE 40:7, PE 40:4, PC 36:6, PC 36:3, PC 36:2, PC 38:4, PC 40:7. The initial evidence suggests that tissues might be classifiable based on the complex lipid acyl makeup, although missing data currently hampers the building of such a model.

Organ wide acyl modification again showed a relatively consistent consensus with regards to the response to the two diets (Appendix A), although tissue-specific dietary effects may also be alluded to by, for example, differences in the magnitude of the lipid response, and the counter-trend of certain lipids in specific tissues. Those effects, which run contrary to the organ wide consensus, or differ markedly in magnitude in either percentage or fold terms can be observed within PC 42:10, PC 36:4, PC 36:2, PC 36:6, PC 32:1, PE 42:6, PE 38:6, and PE 36:2. Whilst percentage composition data and fold data generally tend to align, there are deviations whereby counter-trends are more readily identified. On the other hand, correlations that are similar and strongly reflected in both organs and timepoints are most clearly visible in lipids such as PC 36:3, PC 36:2, PC 40:8, and PC 40:7, although even these lipids demonstrate outliers with regards to acyl composition within a certain tissue, for example. With regards to temporal patterns, which may demonstrate a strengthening or weakening of a fold or percentile change over time, consistent patterns are difficult to determine. However, within the initial QTOF data, several consistent temporal patterns, either strengthening or weakening of fold changes, could be observed across time, as illustrated by PE 36:4, PE 38:4, PE 40:9, PE 42:9, PC 32:1, PC 40:7, PC 40:8, PC 40:9 and PC 42:10. Acyl data demonstrated fewer of these temporal trends, with the majority being more evident within percentage composition data within lipids, such as sn positional isomers PC 16:1/20:5(b), PC 20:4/20:5(a), PC 18:3/22:6, and all those acyl isomers comprising PC 42:10. It was found that there was some overlap between the datasets, for example, PC 40:9 and PC 42:10, indicating that there may be both proportional and absolute temporal trends in effect within a specific complex lipid, i.e., PC 42:10.

A key requirement of the reverse phase method was to resolve what are thought to be sn positional acyl isomers. Positional isomers were denoted as such based on their fragmentation spectra, however, their sn designation cannot be confidently attributed and therefore isomers were designated as either a or b, or c if more than two were detected, and are taken as a ratio of one isomer over the other. Surprisingly, several putative positional isomers were detected (Figure 9) and, in one lipid, PC 32:1, four isomers were detected. Where representative numbers of tissues were found to contain these isomers, certain isomeric trends appear to be influenced by diet, albeit not all statistically valid. This was highlighted most prominently within PC 16:0/20:2, PC 16:0/20:3, PC 16:0/20:4, and PC 20:4/22:6, where one isomer appears favoured in salmon fed either the control or GM diets. Again, there appeared to be tissue-specific variations with regards to either magnitude, for example, eye, when compared with intestine and liver for PC 20:4/22:6, and direction, as seen with the brain (intermediate timepoint) in PC 18:1/22:6, which may be characteristic of the organ. In many instances, only fatty acyl isomers were detected as opposed to putative sn isomers, again indicating a potential tissue or temporal discriminating factor. However, owing to the uncertainty of the isomeric identification, further elucidation of lipid biosynthesis and its potential alteration by diet was difficult to elaborate on.

## 4. Discussion

### 4.1. Lipid Class Trends

Previous studies have found that tissue fatty acid compositions generally reflect the diet, and diets containing elevated levels of LC-PUFA such as EPA and DHA generally tend to promote LC-PUFA enrichment within tissues [24,25,26], and the subsequent elevation of complex lipids containing these fatty acids [22] with the opposite being true for diets containing mostly mono- and di-unsaturated fatty acids derived from terrestrial sources. In the present study, several tissues at two different timepoints were investigated to establish whether a universal trend exists in fish fed different diets, or whether tissue and temporal variations exist. For the five tissues studied, general trends were apparent across all tissues and timepoints, although exceptions were common and illustrated that certain tissues appear more plastic than others. Overall, a general trend was established, that a diet enriched in the LC-PUFA, EPA, and DHA, as well as 18:3n-3, was more likely to contain elevated levels of lipids containing larger carbon numbers, such as 42 and 44 carbons, with the inverse being the case for an 18:1-rich diet, which showed an elevation in 32, 34 and 36 carbon-containing lipids, corresponding to lipids such as PC 32:1, 32:2. 34:1 and 34:2. With regards to the unsaturation level, the LC-PUFA enriched diet resulted in increased levels of highly unsaturated complex lipids, containing 8, 9, and 10 double bonds, though lipids containing 4 double bonds were also more prevalent, owing to the inclusion of acyl chains likely derived from 18:3n-3, as illustrated by acyl chains containing 3 and 4 double bonds being incorporated into complex lipids. The control diet generally resulted in mono- and di-unsaturated complex lipids being enhanced within tissues, although lipids containing 6 or 7 desaturations also appeared to increase on average, with examples such as PC 36:6 and PC 40:7. Regarding acyl chains, the LC-PUFA enriched diet appeared to favour acyl configurations of 20–22 carbons, containing 3–4 double bonds, with 5–6 double bonds being marginally increased by the diet. Acyl configurations which were increased with the control diet generally contained 16 carbons, and 1 double bond, though 18 carbon acyl chain inclusion preference could not be determined conclusively owing to their high prevalence in both diets as 18:1 and 18:3n-3.

### 4.2. Tissue Plasticity

With regards to the amplitude with which lipid modulation occurred, the tissue with the lowest plasticity was the brain, followed by the eye to some extent, which generally corroborates previous work [24,25,26]. The brain generally exhibited a limit on the fold change induced by diet, being the most limited in range compared with other tissues. This limitation in response was highlighted by the lack of meaningful modulation by the control diet, suggesting a potential homeostatic process by which certain fatty acids cannot be included beyond a certain threshold, as evidenced by the responsiveness of other tissues to the control diet while allowing enhancement of longer and more unsaturated lipids if these lipids are elevated within the diet. Previously, the response of the brain to diet, particularly to vegetable oil and fish oil, was contrary to this, with DHA within the brain appearing unresponsive to dietary DHA level, though this was only examined for the total pool of DHA and 18:2n-6, with the latter only comprising less than 2% of total fatty acids, though appeared to be modulated by a diet rich in 18:2n-6 [27]. Within rats, however, a small increase in total DHA within brain tissue was observed with increased dietary DHA, though 18:1n-9 and 18:2n-6 within the brain showed no obvious response to dietary levels of these fatty acids [28]. A similar, yet more pronounced effect was found within PE in rat brains, with increased DHA in the diet benefiting both the total DHA and multiple diacyl and ether lipids, with 18:1n-9 and 18:2n-6 remaining relatively unchanged [29]. It appears then, that increasing dietary LC-PUFA has a small, but beneficial role in the brain, resulting consistently in increased levels of lipids containing 8 or 9 double bonds, with 4 and 5 double bond-containing lipids also elevated. This increase appeared to occur across the entire range of carbon chain lengths, and for the majority of lipid classes. Glycosylated lipids also appeared to be responsive to higher levels of polyunsaturated lipids within the diet, with a countertrend being observed in the eye. These trends may be driven by 18:3n-3 and its attachment to the long-chain base of the glycosylated sphingolipid, whereas eye glycolipids appear driven by mono- and di-unsaturated fatty acids. Interestingly there appeared to be little crossover between the compounds identified within the two tissues, indicating that both the brain and eye may benefit from tailored unsaturation profiles that are unique to their function. In humans, the functionality of the lens appeared to be due to the unique membrane structure, high degree of saturation, and high levels of dihydrosphingomyelin [30], despite glycosylated sphingolipids accounting for less than 1% of the total retinal lipids, with cerebrosides constituting a greater proportion in brain tissue, predominantly within the myelin sheath [31].

Gill, on the other hand, appeared in direct contrast to the brain, being predominantly modulated by the control diet. This appeared to occur across the majority of carbon numbers, within the two major lipid classes PC and PE, and, in those lipids containing 8 or fewer double bonds (other than those with 4), although PE showed a relatively small number of lipid modulations as demonstrated by the lower number of fold changes detected. However, gill was found to respond in line for key lipids, which was especially noticeable within PC. Certain highly unsaturated lipids such as PC 40:8 and 40:9 were increased by the increase in total polyunsaturates found within the GM diet, as did intermediate lipids such as PC 36:4 and 38:4, driven by 20:4 and 22:4. The most plastic and variable organs observed were the intestine and liver, which was unsurprising owing to their roles in both digestion and lipid modification, and their diverse enzymatic activity and general role in lipid modification and export to other tissues. Therefore, the greater fold change variance, compared with relatively static tissues, such as the brain and eye, appeared to be a consequence of their function, as well as the propensity of tissues such as the liver and intestine to autolyse and continue to modify metabolites post-harvest.

### 4.3. Acyl Complexity and Counter Trends

The primary goal was to establish whether isomeric effects could be discriminated against and whether this additional level of detail would help to further our understanding of the impact of diet and its complex effects on lipid metabolism. What was illustrated was the diverse acyl variety present in tissues, which generally reflected the diet, although significant tissue and temporal variations existed that may provide an opportunity to both classify and understand the alterations in lipid metabolism in finer detail. With the general tendency for the majority of tissues to follow a certain trend, as illustrated with the two diets tested, those trends which run counter to the rest may provide insight into the tissue or temporal lipid regulation. The most obvious counterpoints arose within the brain and gill, though smaller deviations from the consensus were also present. Some notable exceptions include variations in abundance, whilst still maintaining comparable fold changes, distinguishing brain, eye, and gill, for example. These were evident in PE 36:2, PE 40:7, PE O-40:8, PC 36:4, PC 36:6, PC 40:8 and PC 40:9 as examples. The altered abundance profiles, as measured by peak intensity, provided a measure of the prevalence of these lipids, and that abundance or deficits of specific lipids may be markers of certain tissues and timepoints. Certain lipid types also effectively discriminate both brain and eye by their presence and are also altered by diet. These lipids constituted the ether lipids (plasmalogens) and the glycosylated lipids, bound both as sphingolipids and MGDG. Ideally, tissues and timepoints may be distinguished through multivariate analysis. However, owing to the initial untargeted approach and subsequent refinement of these detected lipids, the large number of missing values between different tissues and timepoints makes such modelling difficult, and likely to be swayed by the absent data. Understanding the lipid flux within individual tissues, and its alteration by diet is likely a complicated endeavour, more so than in metabolomics, where specific pathways for individual metabolites have been established. Within lipids, although pathways such as the Kennedy, Lands cycle, fatty acid biosynthesis, and lipid class synthesis have been established, the fact that these pathways act, or have an impact, upon multiple substrates, be they biogenic or diet-derived, generally results in lipids being grouped into their classes, rather than individual metabolites, potentially obfuscating key links between lipids. Such finer grain lipid metabolism, however, can be deconvoluted through flux analysis, usually through stable isotope analysis [32]. This may go some way to understanding the complex interplay between the myriad of individual lipids, rather than the current broad lipid class models.

On studying the effect of growth/age, and whether differences could be determined which might indicate the age of the fish and/or whether lipid effects are strengthened or weakened over time, there were instances, predominantly in the initial screening data, indicating that fold or proportional strengthening and weakening effects might occur over time. How these trends relate, and whether they are the result of lipid flux as the fish increases fat storage and reduces bodily growth, for example, is currently difficult to determine due to the limited number of observable trends. The correlation between timepoint data, specifically the magnitude shift of the fold change or percentage composition over time, might enable individual pools to be connected that, in turn, might help to delineate whether lipids are biogenic in origin or dietarily derived. It is also suspected that different timepoints may be distinguishable and therefore classifiable, owing to differences in individual lipids in either fold or percentage composition terms, and that these different timepoints may eventually be linked to differences in growth stage, sexual maturity, and water temperature, though attributing lipid alterations to these factors is currently not possible and requires further investigation. This classification, however, would require further validation through the targeted acquisition of lipids across all tissues and timepoints.

The detection of acyl isomeric differences in relation to diet is primarily attributable to the fatty acids found within the diet and are likely due to the digestion of fatty acids from TAG, the main lipid class in aquafeeds. However, acyl isomeric and putative sn isomers could also be due to the configuration of TAG molecules. Whilst it is difficult to directly determine an association, owing to the lack of definitive sn positional data, the fact that isomeric trends exist indicates that further structural analysis is warranted. The mechanism by which acyl configuration is maintained is likely through monoacylglycerols (MAG) after lipase digestion within the gut, with 2-MAG the predominant pancreatic lipase product [33] taken up across the enterocyte membrane. It is these 2-MAGs that possess a structural fingerprint that can be built upon, with various acyl-CoAs derived from free fatty acids then used by monoacyl- and diacylglycerol-acyltransferases to reconstruct TAG, or through phospholipid-specific pathways. While free fatty acids cannot transfer a structural signal, if deficient in LC-PUFA, elongation of dietary and de novo fatty acids may carry with it a specific structural signal, for example, by biosynthesis of DHA through Sprecher’s shunt, which may be distinct from that obtained from direct incorporation of LC-PUFA into complex lipids. The former, that isomeric signals derive from structured TAG, is predicated on the fact that structural alterations exist within lipids used for aquafeeds. It has been found that the positional distribution of specific acyl groups can vary, with both genetically engineered *C. sativa* and *B. napus* demonstrating a strong preference for DHA incorporation into the sn-1/3 position in TAG [9,17], while EPA was either incorporated to a greater extent into the sn-1/3 position, or was equally distributed between the positions [34], but was enriched in the sn-2 position in PC. The distribution of DHA appears similar to that found within certain single-cell oils [18]. Fish body oil, however, was found to have DHA preferentially esterified (80%) to the sn-2 position, whereas EPA was split equally between sn-2 and sn-1/3 positions in TAG [13]. In contrast, PC was found to contain EPA and DHA preferentially bound (77%) to the sn-2 position. DHA incorporation in TAG has also been corroborated in anchovy oil, with DHA 1.7 times as likely to be found in the sn-2 position, though EPA was found to be 4.3 times as likely to be found in either the sn-1 or -3 positions [14]. In general, fish oils appear to favour DHA in the sn-2 position, whereas EPA is generally found in the sn-1/3 position [15,16]. Differences in positioning might also be observed in 18:2n-6 and 18:3n-3 and their subsequent elongation products, as both these fatty acids are present in both fish and terrestrial plant oils, albeit 18:3n-3 is difficult to resolve in ^13^C NMR and, depending on the plant and cultivar, might be limited in concentration. This relative positioning of EPA and DHA within TAG has also been found to be beneficial to serum and liver cholesterol and TAG concentrations [35], though more so in the sn-2 position for DHA, and in the sn-1/3 position for EPA, with EPA in this position resulting in modulation of the eicosanoids PGI_2_ and TXA_2_. This indicates that certain structural forms of lipids, specifically TAG, which are predominantly found within feed formulations, may have some impact on fish health, or have implications for lipid biosynthesis within the fish itself, with these biochemical modulations initially appearing to be attuned to the regiospecific configuration of fish oils. The initial evidence that modified plant oils do in fact deviate from standard fish oils, specifically in DHA positioning within TAG, lends credence to the fact that isomeric signals may be transmitted from the diet to organs, with this signal being found within the putative sn isomeric ratios. The detection of these acyl isomers provides a valuable resource for the development of more targeted approaches, with the aim of filling in data gaps resultant from an untargeted approach and exploring those lipids that display alternative acyl configurations, as well as altered fold and compositional responses to diet.

With regards to assessing these isomeric alterations routinely, targeted approaches utilizing multiple reaction monitoring (MRM) are potentially the methods of choice, although traditional HILIC and reverse phase methodologies cannot reliably separate the myriad of acyl isomers that constitute the lipidome. Whilst certain columns such as charged surface hybrid (CSH) [36] are known to separate isomers based on double bond configuration, the thorough assessment of isomers utilising isocratically derived methods is slow and time-consuming, usually requiring separate methods for both polar and neutral lipids. This is generally impractical, though likely to become easier with the increased adoption of ion mobility. The determination of what are putatively identified as sn positional isomers, and the identification of trends associated with one isomer over another, for example, PC 20:4/22:6 and PC 16:0/20:4, as well as the detection of four potential isomers of PC 16:0/16:1, indicates that there may be a signal which can be transferred from the diet or derives from the biogenesis of specific lipids under certain conditions, such as LC-PUFA scarcity. It is also worth noting that not all tissues and timepoints were found to contain these putative sn isomers, only being found to contain fatty acyl rather than positional isomers. This may be due to low abundances and the inability to consistently resolve closely eluting isomers, although it may also be a characteristic feature of the tissue or timepoint.

## 5. Conclusions

Overall, it was found that phospholipid molecular species in salmon tissues generally reflected the given diet, though variations were found in the response to the diets, specifically in the brain and gill, indicating some need for homeostatic regulation of certain fatty acids. Tissue-specific variations also existed within certain lipid classes, such as the glycosylated lipids, including alterations in abundance, fold amplitude, lipid acyl configurations as well as putative sn isomers, across tissues and timepoints, indicating the existence of unique organ and temporal lipid signatures, which both defined the tissue and suggested differences in their lipid biosynthetic pathways. The presence of putative sn isomers highlighted the need to explore the interplay between novel diets and organ biochemistry in more detail, with structurally different lipids derived from synthetic biology potentially capable of influencing the structure of organ lipids that, in turn, may have consequences for processes such as eicosanoid formation.

## Figures and Tables

**Figure 1 metabolites-12-00851-f001:**
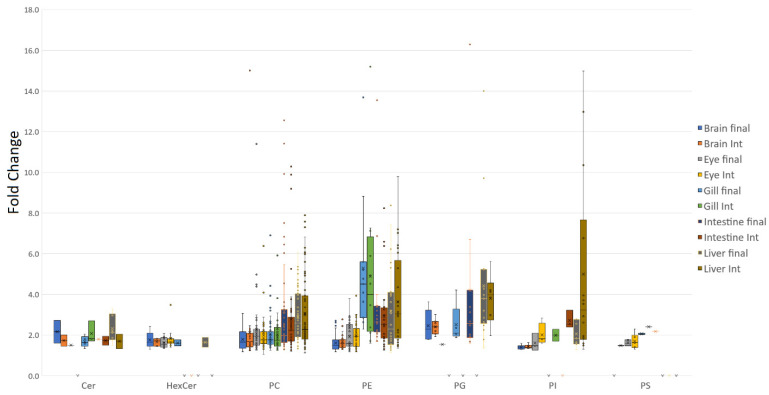
Phospholipid class fold distribution for untargeted QTOF data. The average fold distribution of lipids determined to be significant showed that, in general, the brain and eye exhibited smaller fold changes across the majority of lipid classes. The liver, however, generally exhibited the greatest variance in lipid alteration in response to diet.

**Figure 2 metabolites-12-00851-f002:**
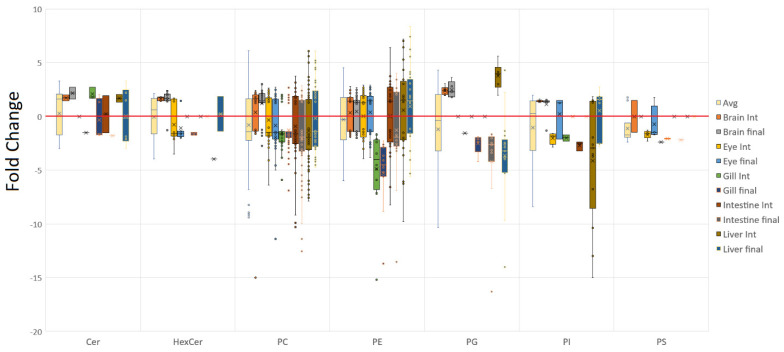
Phospholipid bias toward GM or control diets. Positive values indicate a general trend of being increased in fish fed the GM diet, whereas negative values indicate a general trend of being reduced by the GM diet, compared to fish fed the control diet. The brain in general was found to be positively affected by the GM diet, whereas the opposite was the case in the gill with the major lipid phospholipid class PC being primarily increased in fish fed the control diet.

**Figure 3 metabolites-12-00851-f003:**
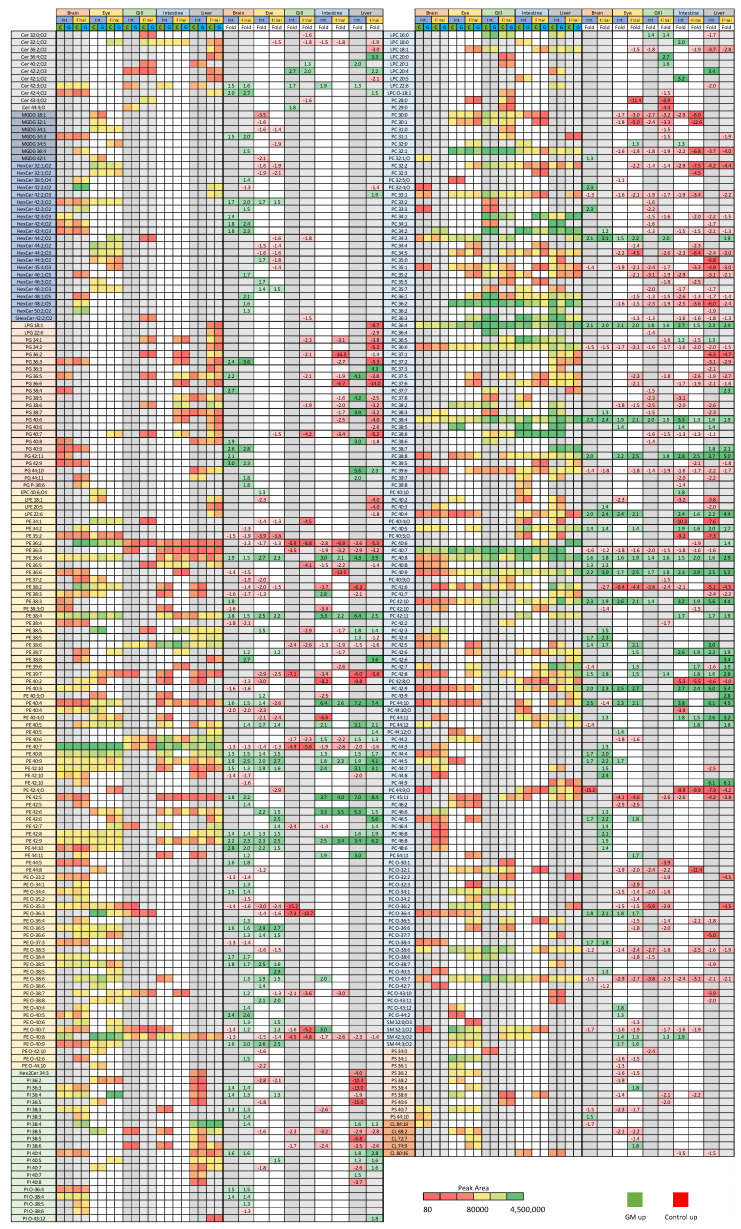
Lipids found to be significantly elevated by either the control or GM diet using the untargeted SFC-MS/MS methodology. Lipids were determined via ANOVA and OPLS-DA analysis and formed the basis for the semi-targeted reverse phase LC-MS/MS analysis. Data are given as all ion normalised peak areas (with colour scale for reference), and fold change data, with green representing elevated levels in the GM diet vs. control, and red representing elevated levels in the control diet vs. GM.

**Figure 4 metabolites-12-00851-f004:**
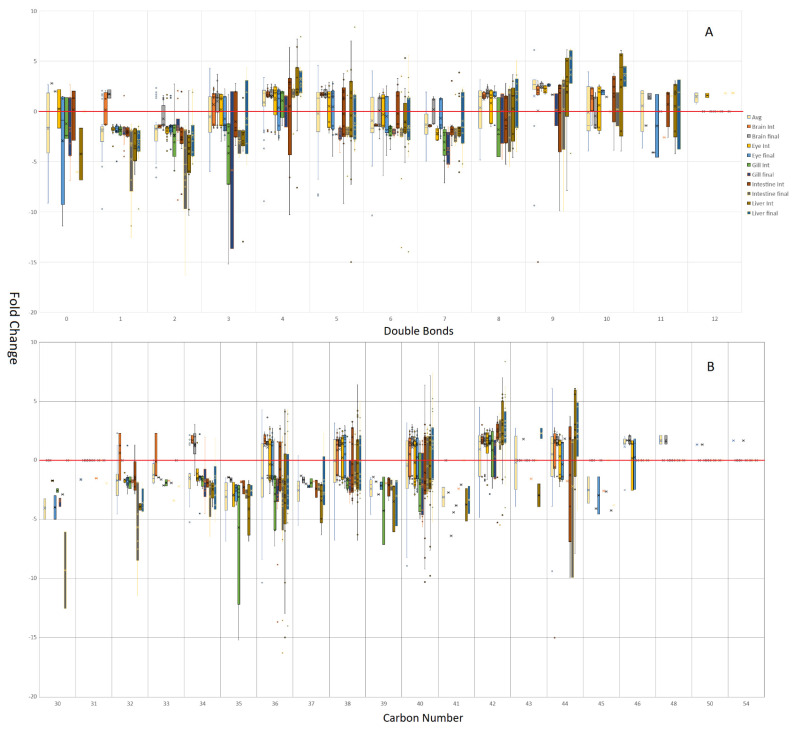
Lipid double bond (**A**) and carbon number (**B**) bias toward the GM or control diets, with positive values indicating a general trend of being increased by the GM diet, whereas negative values indicate a general trend of being decreased by the GM diet compared to fish fed the control diet. The brain was consistently found to have the majority of lipids increased by the GM diet, in contrast to gill, which was biased by being more likely to show lipids increased by the control diet. On average, lipids containing 1, 2, and 7 double bonds, and comprised of 32, 34, and 36 carbons, were more likely to be increased by the control diet, whereas lipids containing 4, 9, and 10 double bonds and comprised of 42 carbons, were more likely to be increased by the GM diet.

**Figure 5 metabolites-12-00851-f005:**
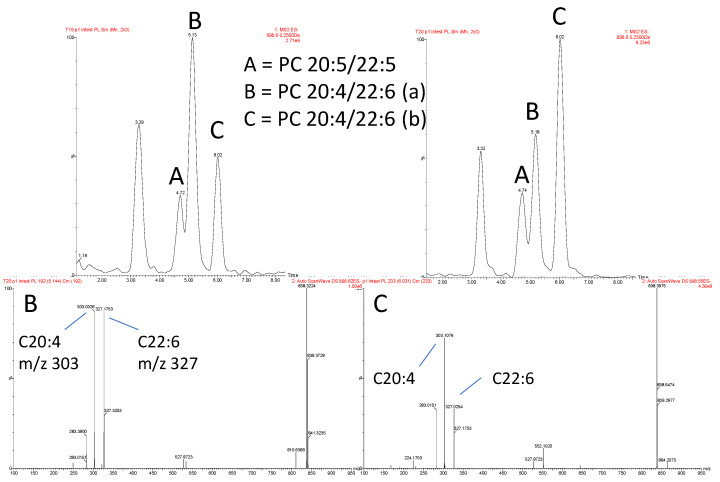
Acyl and putative sn isomer separation based on BEH C18 reverse phase analysis. Putative identification of sn isomers was made based on unit mass and CID fragmentation of precursor ions in negative ion mode. In this example, PC 42:10 is comprised of 3 isomers, with a shift in the relative proportion of the two putative sn isomers in response to diet. (a) and (b) nomenclature is used to designate the putative sn isomers.

**Figure 6 metabolites-12-00851-f006:**
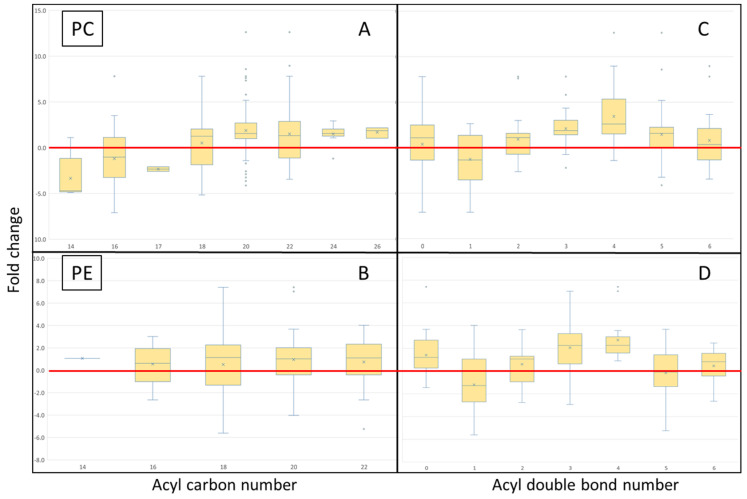
Averaged individual PC and PE bias plots for all tissues and timepoints for both acyl carbon number and number of double bonds, expressed in fold change. (**A**) acyl carbon number in PC, (**B**) acyl carbon number in PE, (**C**) double bond number in PC, and (**D**) double bond number in PE. The GM diet preferentially increased lipids containing acyl chains with 3 and 4 double bonds, comprising 20 and 22 carbons. The control diet preferentially increased lipids containing acyl chains containing 1 double bond, comprising 14 and 16 carbons.

**Figure 7 metabolites-12-00851-f007:**
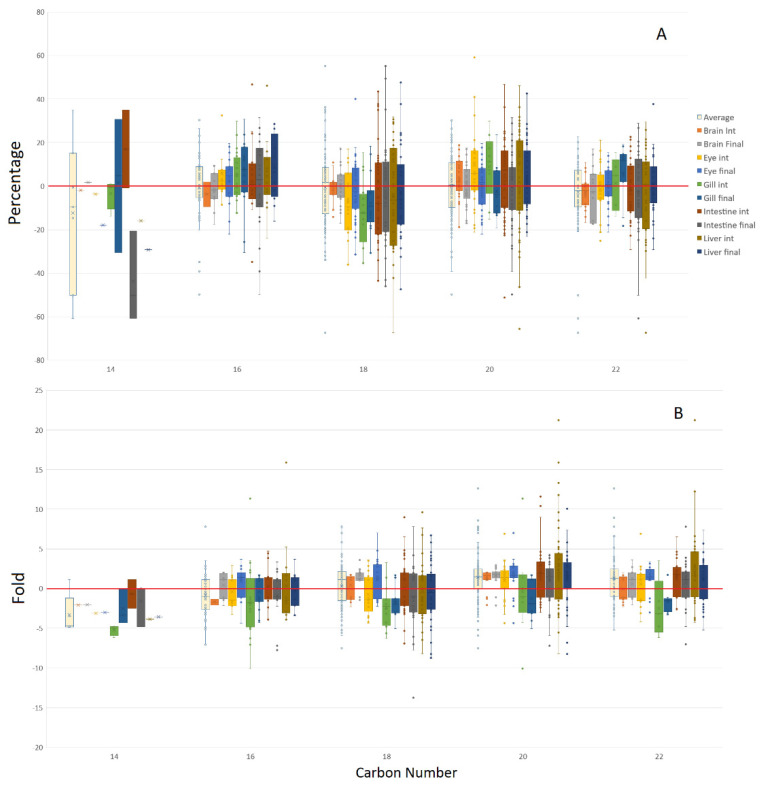
Acyl lipid carbon number bias toward the GM or control diets for combined PC and PE lipid pools represented as a percentage (**A**) or fold data (**B**), with positive values GM biased, and negative values control biased.

**Figure 8 metabolites-12-00851-f008:**
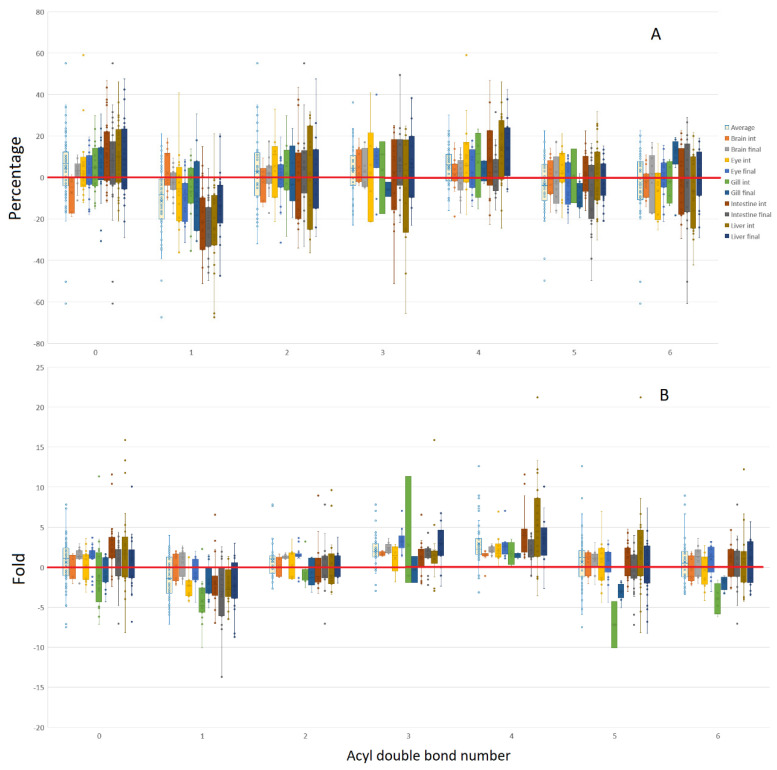
Acyl lipid double bond number bias toward the GM or control diets for combined PC and PE lipid pools, represented as a percentage (**A**) or fold data (**B**), with positive values GM biased, and negative values control biased. On average, acyl groups containing 1 double bond were increased by the control diet, whereas those containing 3–4 double bonds, tended to be increased by the GM diet.

**Figure 9 metabolites-12-00851-f009:**
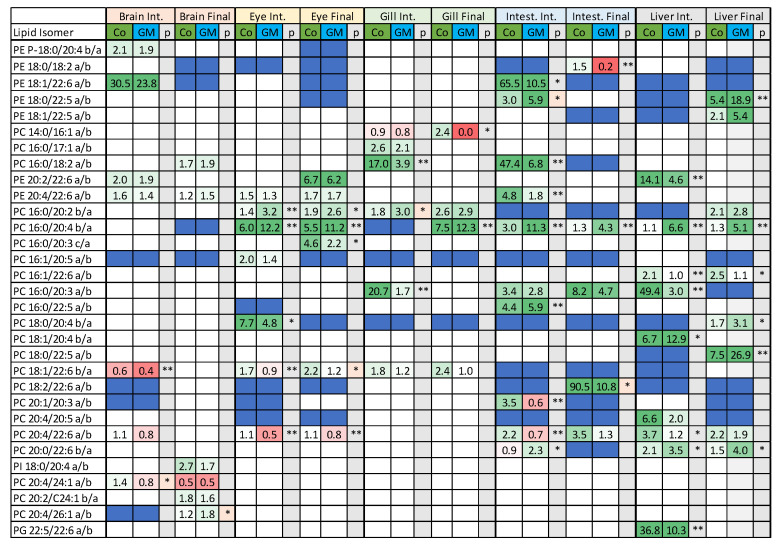
Potential sn isomers identified using the reverse phase LC-MS/MS methodology. Data are presented as the ratio of one isomer over the second isomer, with ratios determined using peak area data. Blue squares indicate that sn positional isomers were not detected, however fatty acyl isomers were detected. * *p* = 0.01–0.05, ** *p* < 0.01, red boxes indicate *p* values fell below the Benjamini-Hochberg critical value.

**Table 1 metabolites-12-00851-t001:** Diet formulation and fatty acid composition of the control and GM diets. The provenances of the ingredients used were ^a^ Biomar AS, Brande, Denmark, ^b^ Rothamsted Research, Harpenden, UK, ^c^ DSM, Basel, Switzerland.

	Control	GM
*Feed ingredients (%)*		
Fishmeal ^a^	7.5	7.5
Vegetable protein ^a^	36.1	36.1
Land animal protein ^a^	10.0	10.0
Wheat ^a^	11.2	11.2
Fish oil ^a^	10.8	-
Rapeseed oil ^a^	20.8	-
Camelina oil (transgenic) ^b^	-	31.6
Premix ^c^	3.6	3.6
Yttrium oxide ^c^	0.005	0.005
*Fatty acid profile (%)*		
Total saturated^1^	18.1	14.7
Total monoenes^2^	49.1	22.2
18:2n-6	15.2	19.6
18:3n-6	0.1	1.5
20:4n-6	0.4	1.7
Total n-6 PUFA	16.0	25.2
18:3n-3	5.8	19.1
20:5n-3	4.8	5.7
22:5n-3	0.7	3.8
22:6n-3	2.4	5.3
Total n-3 PUFA	15.1	37.7
*MOPA*		
Moisture %	6	6
Energy–crude (MJ/Kg)	26	26
Protein–crude (%)	36	36
Fat–crude (%)	36	36
Ash (%)	4.2	4.2

**Table 2 metabolites-12-00851-t002:** Reverse phase gradient conditions.

	Solvent			
Time	A	B	C	D
0	100	0	0	0
5	100	0	0	0
5.1	0	100	0	0
18.5	0	100	0	0
35	0	50.2	49.8	0
42	0	20	80	0
42.1	35	0	0	65
52	35	0	0	65
56	100	0	0	0
61	100	0	0	0

## Data Availability

Data is contained within the article and Appendix A, as listed previously.

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
