# Peer review of "Profiling Phospholipids within Atlantic Salmon Salmo salar with Regards to a Novel Terrestrial Omega-3 Oil Source"

_metabolites, 2022, doi:10.3390/metabo12090851_

Round 1

Reviewer 1 Report

the ms deals with actual topic and generally the ms is well-prepared, it can be considered for publication

Author Response

We'd like to thank the reviewer for taking the time to review our paper and their overall positive assessment of our work. Many thanks.

Reviewer 2 Report

Manuscript ID: metabolites-1876436

Profiling Phospholipids within Atlantic salmon Salmo salar with Regards to a Novel Terrestrial Omega-3 Oil Source

Reviewer comments:

An interesting manuscript and with novelty in the  screening approach on identifying and characterize lipids and lipid isomers and their differences in salmon tissue by providing LC-PUFAs through a standard diet or included in a GM diet. The paper is well written although lipid analytical methodology  and data treatment at some part slightly complicated to follow..

A remark related to the reason for presenting data on two time timepoints for samplings, when not commented on in terms of lipid metabolism in relation to composition, fish growth, size, season, fecundity ect. or any other given hypothesis in the  introduction ? 

Final fish weight was some avg 1.9 kg = more than 10 times weight increase from start of the dietary treatments. Authors do not provide info. about fish size at the intermediate sampling period after some 23 weeks ? , although I reckon growth and fish size at this point was  more than sufficient to conclude on such data with no further temporal patterns necessary ?

As the two timepoints seems difficult to justify in relation to be able to interpret on effects on growth/ age (discussion line 597.. an onwards),-   I wonder if data from the two timepoints could have been successfully merged to strengthen statistics and validation of the methodology applied as some of the data indicate quite some variations ?

When fish were finally harvested in March do you consider any influence on results  related to matureness (any analysed proximate composition etc) , that may interfere with results from the intermediate period ?

Fish, were they all females or a mixed population, and if so do you have info on sex of the sampled fish ? 

Introduction:

Line 40-41: Partly correct, but be aware that heterotrophic oils (i.e.Corbion: Alga Prime product) are already used at commercial scale /by large aquaculture feed companies:

Makers of algae-based fish feed ingredient celebrate reaching 350,000t mark - Undercurrent News

Mat & methods:

Line 110-111: Is this protein content correct: seems very low only 30 % ? – normal commercial feeds for this size of Atlantic salmon is some 37-41 % protein (as fed)

Table 1. Please provide information about origin of source of ingredients  and the analyzed nutritional composition (protein, lipid, energy etc.)

Line 121: or at line 115 Please provide info of the size of the sampled fish for each treatment group (to avoid any influence of size)

Line 121-122: … “and  six biological replicates per diet (n-6)” ??  you mean 3 replicates per diet (= a total of 6 netpens)

Lines 185-209: (especially line 195-203): These seem quite complex analytical procedures ( could you add a ref. on this methodology). Any potential possible errors in the analytical procedure.  

Discussion:

Nicely written.

You could add a few remarks owing to the reason for having the two timepoints in relation to the statements and discussion line 697-609,-  and my initial remark, when not able to conlude on this influence.

Miscellaneous:

Line 108: here you use startpoint as June 2018 (week 25 ?) while line 115 you write week 48 as intermediate sampling-   please uniform to weeks.

Fig. 1: I guess it is more to see the overall fold change pattern of the different tissues, as it is quite difficult to detect exact values especially SD or SE ? values appear vague  but seems very high for many lipid classes and tissues- any comments on this ?

Fig.2: ..  negative and positive values indicate  trend of being reduced or increased by the GM diet – compared to fish fed the control diet”. ..(and text line 236-241) = there may be some positive or negative trends for some tissues like brain and gilI within certain lipid classes- but with the quite high standard deviations indicated only very few statistical differences  ? also indicated by avg values.

For me it seems that PC had a relatively equal split for most tissues similar to PE ?,

Fig. 3: Very comprehensive – maybe highlight most interesting values in a figure while remaining data could be as a supplementary figure

Alternatively show only final and not intermediate data

Reviewer 3 Report

The SCF-MS/MS and LC-MS/MS were used by the author to determine and characterize lipids in Atlantic salmon (Salmo salar). In my opinion, the author performed admirably in experimental and data analysis. They are also well-written and offer a thorough explanation of all topics. Thus, I would recommend that this article be published in Metabolites.

Author Response

We'd like to thank the reviewer for taking the time to review our paper and their positive assessment of our work. Many thanks.